# Automated Generation of Executable Cross-Language Background Knowledge

## Motivation

Certain problems require specific semantic knowledge to be solved, e.g. the transformation "January → 1". Without the required domain specific knowledge, a program synthesizer can't generalize towards a program that also learns the other months of the year. Online repositories of code contain many of these transformations and they can be used to improve the semantic strength of a synthesis technique [He et al., 2018]. By reusing previously written code that contains parts of or the entire transformation, these domain specific problems can now be solved as well. This background knowledge, in the form of method declarations, is however not easily available in a uniform format. The lack of formalization of this process results in handcrafted crawling and parsing software for each program synthesis technique, as can be found in recent works [He et al., 2018, Yan and He, 2018]. Moreover, when the program space spans over multiple programming languages these techniques have to be adapted by the synthesis researchers for each separate language. This work tries to formalize the process of generating executable background knowledge in multiple programming languages that can be used in different synthesis techniques. Such framework would lower the amount of time spent on crawling, parsing and runtime environments within future research projects.

## Framework

The creation of such a background knowledge consists of three main phases: *crawling code*, *selecting relevant functions* and *creating a runtime environment*. Crawling code on a website like GitHub provides a rich and diverse collection of functions for each programming language. The found source code files can be scanned from which method declarations[1] can be collected and filtered according to a set of chosen criteria. Given some runtime environment, these previously collected methods can be executed on provided input examples. An important note is that each of these steps is intertwined with the used programming language. For example, the filtering step requires an AST-like representation of the method declarations to properly process each entry. The libraries used to construct these ASTs are clearly language specific and can't be generalized. These language specific steps have to be clearly marked so that an open source contributor can extend the framework easily by implementing these steps for the new programming language. This reduces the complexity of adding a new programming language to implementing the language specific steps. A collection of function handles processed by this framework, potentially containing multiple programming languages, can thus be ran by merely supplying input parameters, see figure 1. This process severely reduces the amount of software engineering work needed from the synthesis researcher.

Using one uniform framework which clearly marks the language specific steps, has two clear advantages. Firstly, a uniform open source framework only requires a programming language to be added once. Secondly clearly marking the language specific steps, reduces the complexity of extending the framework with another programming language to providing an implementation of the language specific steps. The next subsections will give a high level overview of the current design of the framework which as of writing this paper has implementations for `Python` and `Java`, for a `Prolog` extension see section *Extending to Prolog*. Figure 2 contains

---

[1]In a language like `Prolog` analogous steps can be followed to collect predicates.

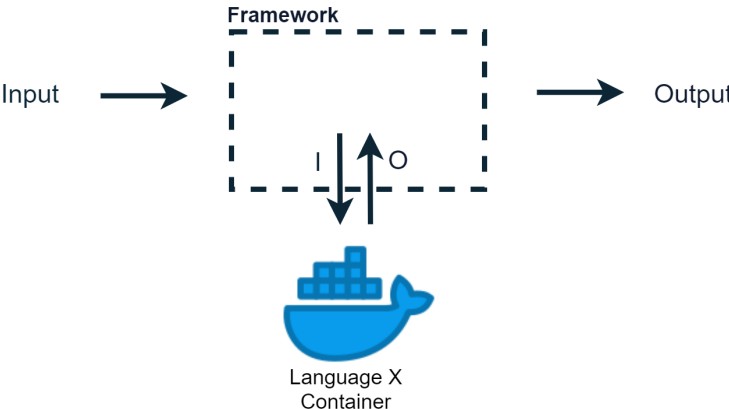

Figure 1: Framework interaction as a synthesis researcher

a high level of overview of the proposed framework. Each of the next subsections describes both the generic part that works for any programming language and the programming language specific part.

### Crawling

External information is needed to provide the aforementioned building blocks. While the source of information can be any collection of readable source files, the implementation of the current framework only supports the GitHub API.

*Abstractions.* Given the description of the desired files, usually a file extension, the process of downloading the code from GitHub and assembling the desired documents is entirely automated.

*Language Specific.* Crawling from more complex data sources or from e.g. `readme` files can be implemented to increase the method declaration yield at a programming language specific level.

### Collecting & Filtering

The found source files could contain methods that provide domain knowledge needed to solve certain transformation tasks. This step collects these method declarations and subjects them to filters so that low quality methods are left out.

*Abstractions.* This phase mainly consists of a collection of filters that reduce the method declaration set. While the abstractions made here will orchestrate the execution of the different filters, it does not bring much value on its own. It does however handle storage, serialization of the AST information and marks building blocks with meta information for future use. This informa-

tion can be used as a syntactic bias over the program space, and thus needs to be collected for algorithmic performance. These aforementioned steps are another example of tedious software engineering work that should not be reinvented every research project.

*Language Specific.* Collecting the method declarations from the found files can be achieved through language specific AST software. This step has to be implemented once for each programming language, but can rely on a vast collection of AST parsers. The most primitive implementation would support functions that operate without their scope-bound context, yet more advanced collection techniques can be added to improve the quality of this step. Some of the filters are also language specific, but as stated before these only have to be written once. For example one of the already implemented `Java` filters will check if the provided code compiles. This does not need to be implemented again and works out of the box. Given the open source nature of this framework, frequently used filters should appear if the framework is adopted within the community.

### Runtime Environment

To run one of the left over methods, it is important that a runtime environment is present that can handle its programming language. The same issue occurs when a filter needs language specific elements like access to a compiler. A true cross-language research project thus requires a runtime environment for each programming language. It would be inefficient to reimplement this for every program synthesis technique as a generic solution would only require one implementation for each programming language. This framework uses Docker to provide runtime en-

**Framework**

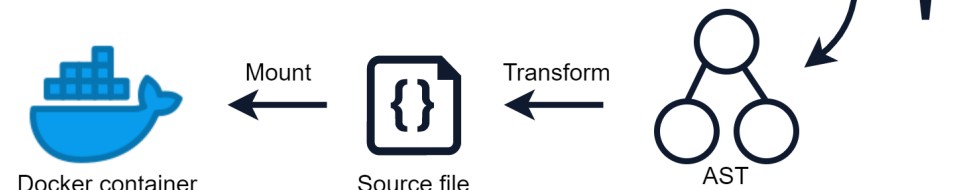

Figure 2: High level overview of the framework.

vironments, as it's widely used and provides an easy installation format for end users.

*Abstractions.* A generic program is written to handle the life cycle of any docker container and retrieves the output of the ran program. Methods that would run infinitely or throw errors will also be caught using the aforementioned life cycle management. Given that any language outputs a JSON, the storage of the output of these containers also can be automated regardless of the used programming language.

*Language Specific.* A docker image must be selected for the new programming language and some JSON serializer software has to be added to the image. The files that should be ran are volume mounted into the container and ran through a run script. From this point on the generic docker wrapper can operate on its own. See the next chapter for an example.

## Extending To Prolog

To show what it takes to extend such a framework, this section will describe the required steps to add `Prolog`. This would also be the running example presented in the workshop.

### Crawling

By adding the "`Prolog`" keyword to the GitHub API and providing the correct file extension, i.e. `.pl`, the framework automatically starts downloading the N most starred `Prolog` repositories on GitHub. These files will contain the predi-

cates that can be used as background knowledge in other synthesis techniques.

### Collecting & Filtering

The predicates from the aforementioned `Prolog` files have to be extracted. A very simple version of this step could make use of a simple regex matching the predicate structure or make us of *prolog_xref*[2] to analyze the contents of the source files. An example of a filter would be an attempt to execute this predicate without its surrounding context, so that non executable predicates within this setup are thrown away.

This rudimentary implementation does not work for predicates requiring dependencies, but it is clear that on a programming language specific level this can be extended to increase predicate yield.

### Runtime Environment

The `swipl` image provides the needed environment for `Prolog` predicates to be evaluated. Within this environment a template file is needed to handle the process of running arbitrary Prolog predicates. This template includes accessing the JSON serializer[3], logging this JSON to `stdout` and wrapping the end user provided input in the *Prolog* file. Once such a template exist, the found predicates can be pasted into this template and ran from the terminal. From there on the generic Docker wrapper can process the output data.

---

[2]https://www.swi-prolog.org/pldoc/doc/_SWI_/library/prolog_xref.pl
[3]http/json

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
