# OpenReview forum: "Automated Generation of Executable Cross-Language Background Knowledge"
_NeurIPS.cc/2020/Workshop/CAP — Reject_

### Official Review · AnonReviewer1 · 2020-10-30
**Incomplete submission**

**Rating:** 3
**Confidence:** 4

**Review:**

This paper outlines a data collection framework for collecting programs in many different languages. However, the presentation is merely a to-do list describing various components to be implemented.

Though this paper could make significant impacts once it is ready. The current presentation is quite premature for publication.

---

### Decision · Program_Chairs · 2020-11-02

**Decision:**

Reject

**Comment:**

Unfortunately I have to concur with the reviewer in this instance.
I agree that such a framework would be useful, but the present draft is too high-level to really spur useful workshop discussion.

I don't really think there would be any real disagreement that the things described in the document would be nice to have, so without more concrete plans or some kind of real artifact to point to, I feel I have to recommend rejection.